# Pushing the limits of fairness impossibility: Who's the fairest of them all?

**Brian Hsu**
LinkedIn Corporation
Sunnyvale, CA
bhsu@linkedin.com

**Rahul Mazumder**\*
LinkedIn Corporation, Sunnyvale, CA
(Massachusetts Institute of Technology, Cambridge, MA)
rmazumder@linkedin.com

**Preetam Nandy**
LinkedIn Corporation
Sunnyvale, CA
pnandy@linkedin.com

**Kinjal Basu**
LinkedIn Corporation
Sunnyvale, CA
kbasu@linkedin.com

## Abstract

The impossibility theorem of fairness is a foundational result in the algorithmic fairness literature. It states that outside of special cases, one cannot exactly and simultaneously satisfy all three common and intuitive definitions of fairness - demographic parity, equalized odds, and predictive rate parity. This result has driven most works to focus on solutions for one or two of the metrics. Rather than follow suit, in this paper we present a framework that pushes the limits of the impossibility theorem in order to satisfy all three metrics to the best extent possible. We develop an integer-programming based approach that can yield a certifiably optimal post-processing method for simultaneously satisfying multiple fairness criteria under small violations. We show experiments demonstrating that our post-processor can improve fairness across the different definitions simultaneously with minimal model performance reduction. We also discuss applications of our framework for model selection and fairness explainability, thereby attempting to answer the question: *who's the fairest of them all?*

## 1 Introduction

While fairness in machine learning has received significant attention in recent years, most existing works focus on one of the many fairness criteria [3]. Consequently, practitioners are perhaps left with no choice but to use their best judgment to apply a single fairness criterion. We suspect that the conflicting nature of existing mathematical definitions of fairness might have led to this undesirable practice of narrowing down fairness-related measurement and mitigation to one chosen definition. This is somewhat analogous to the trade-off between precision and recall while evaluating model performance [5]. Instead of choosing one of precision or recall to evaluate the performance of a classification model, practitioners often evaluate the trade-off and choose models that can maintain a certain level of precision while optimizing recall or vice-versa [14]. In this paper, we provide a framework for explicitly addressing such trade-offs among multiple fairness criteria and model performance toward optimal model selection.

One of the most prolific examples of fairness in machine learning arose from the ProPublica recidivism study ([2]), in which a risk assessment tool called COMPAS was found to be biased against black defendants. But beyond the immediate implications in criminal justice, the study also prompted

---

\*This work was done when Rahul Mazumder was a consultant for LinkedIn.

36th Conference on Neural Information Processing Systems (NeurIPS 2022).

more general studies in algorithmic fairness and, in particular, led to a key result highlighted in [10], [16], and [18] which some colloquially refer to as the "impossibility theorem" in fairness [22, 17]. This theorem essentially states that three common definitions of algorithmic fairness - demographic parity [11], equalized odds [16], and predictive parity [3], cannot be simultaneously satisfied outside of pathological situations. Several works following these initial results have therefore focused on satisfying one metric ([16, 23]) or proposing adaptable methods for different metrics ([8, 27]). Yet, while we do not deny the conclusions of the impossibility theorem, we also believe there have not been sufficient efforts to reconcile the conflicting fairness definitions to the best extent possible. We fill this gap in the literature by translating the trade-offs among multiple fairness criteria and model performance into a constrained optimization problem and propose a post-processing methodology for simultaneously achieving approximate fairness in the conflicting definitions simultaneously.

We believe our framework would alleviate the practitioner from making the hard choice in choosing a particular metric. Instead, if they have a partial ordering of importance amongst the metrics (which many possess), our framework would explicitly allow them to evaluate such trade-offs. The application which we highlight in Section 4 discusses these in detail. Overall, we make three main contributions in this work:

1. We design a flexible optimization framework that returns a post-processing score transformation function that can make scores group-wise $\epsilon$-fair along three definitions (demographic parity, equalized odds, and predictive rate parity) simultaneously. This framework can be applied to any binary classifier that produces a continuous score, can be configured for singular or multiple metrics of fairness, and also can account for fairness vs. performance trade-offs in terms of ROC-AUC.

2. We present a novel reformulation of this non-convex optimization problem as a Mixed Integer Linear Program (MILP) [31]. This reformulation allows us to find provably globally optimal solutions. We further show that in practice, we can consistently find better solutions through our global optimization method compared to local optimization methods in a reasonable time.

3. We discuss and extend our framework from a post-processing mechanism to a tool that can aid practitioners in better understanding their data and models' empirical fairness characteristics and trade-offs and compare these traits across models.

The rest of the paper is organized as follows. In Section 2 we mathematically define fairness metrics and the multiple fairness optimization problem. We discuss the optimal solution via MILP in Section 3. We try to answer the question of "who's the fairest of them all?" through our applications and experiments in Section 4. Finally, we conclude with a discussion in Section 5. We wrap up this section with a discussion of the related literature.

**Related Work:** Early works [10] exploring the conflicts between fairness definitions show that for a binary predictor, predictive rate parity conflicts with equalized odds ([16]) unless base rates are equal or the model is perfectly predictive. Chouldechova [10] also considers trade-offs between the fairness definitions in the binary prediction case. Kleinberg et al. [18] generalizes this study by showing that statistical (i.e., demographic) parity is also inconsistent with predictive rate parity and equalized odds. The same paper studies these inconsistencies in a more general bin-wise prediction setting and shows that approximate fairness definitions (predictive rate parity, equalized odds) can simultaneously hold but only under $\epsilon$-approximate equal base rates or $\epsilon$-approximate perfect performance. This work further proves that there is an inherent trade-off between fairness and loss.

Beyond impossibility theorem results, several works have focused on trade-offs between fairness and model performance [16, 12, 27, 32]. They develop in-processing solutions aimed at reducing one metric while maintaining accuracy. A few authors have analyzed trade-offs or attempted to achieve multiple fairness. One example is Pleiss et al. [26], which shows that predictive rate parity and a relaxed form of equalized odds are reconcilable under a randomized prediction scheme. Another notable example is Celis et al. [8], which develops a flexible in-processing approach to achieve multiple types of fairness (potentially at the same time). Like Celis et al. [8], the framework we develop also aims for a flexible approach to focus on one or many fairness metrics simultaneously. However, our method is distinct in that it is a post-processing based solution and also more general as it works for continuous scores (rather than binary classification). Our work is most closely related to Nandy et al. [23] in terms of the underlying score transformation mechanism, and we leverage some of their methods. However, whereas [23] only targets equalized odds, we go further and include

demographic parity and predictive rate parity in our framework—this posits computational challenges, which we address by proposing novel methodology based on integer programming.

As noted at the end of [18], some open questions are how to optimally assign scores to satisfy multiple criteria when base rates are equal and additionally, how to satisfy predictive rate parity and either equal TPR or TNR when one cost outweighs the other. To our knowledge, no prior work has attempted to reconcile all three fairness conditions (demographic parity, equalized odds, and predictive rate parity) simultaneously with model performance through a post-processing framework. Although the post-processing methods tend to be less flexible for fairness-performance trade-offs than their in-processing counterparts, they can be much more easily added on top of any existing model training pipeline. This makes a post-processing approach modular, and in particular, more appealing in complex web-scale recommender systems that use (a combination of) certain prediction scores to rank a list of items.

## 2 Multiple Fairness Optimization

We consider a binary classification problem where the $i$-th observation is characterized by their label $y_i \in \{0, 1\}$, their group membership $g_i \in \mathcal{G}$, and model predicted probability (also known as a risk score [18]) $s_i \in [0, 1]$ for $i = 1, \ldots, N$. The corresponding random variables are denoted as $Y$, $G$ and $S$ respectively. To set up the problem, we discretize the scores into nonempty bins $b \in \mathcal{B} := \{1, \ldots, |\mathcal{B}|\}$ by using, for example, a quantile transformation (we will denote $|\mathcal{B}|$ as $B$). Additionally, let $N_{b+}^{[g]}$ denote the number of group $g$ positive class ($y_i = 1$) instances in bin $b$, $N_b^{[g]}$ denote the total number of instances of group $g$ instances in bin $b$, $N_+^{[g]}$ ($N_-^{[g]}$) be the total number of group $g$ positive (negative) instances, and $N^{[g]}$ be the total number of group $g$ instances. Lastly, our approach seeks to achieve fairness by moving instances from one bin to another bin: hence, we define variable $x_{bb'}^{[g]}$ as the probability of moving an instance of group attribute $g$ and score in bin $b$ into a new bin $b'$ [2]. In other words, for every group $g$, the collection $\{x_{bb'}^{[g]}\}_{b,b'}$ can be represented as a $B \times B$ transition matrix (with additional constraints, as discussed next).

For the optimization framework and the remainder of this paper, we translate a single fairness definition (e.g. equal true positive rate) as a constraint that controls for the worst-case violations across all bins. Below, we discuss different fairness constraints that we consider in our framework, and describe how they can be represented in terms of the optimization variables $\{x_{bb'}^{[g]}\}$.

### 2.1 Fairness Constraints Under Binning Framework

**Demographic Parity (DP):** For simplicity, we assume that there are exactly two groups $g \in \{1, 2\}$ as we formulate the fairness metrics, although everything discussed below can be applied for non-binary groups by using pairwise constraints, which we elaborate on in Section 2. Starting with demographic or statistical parity from [11], this condition states that the model's predicted score is independent of group membership. This is equivalent to

$$P(S = s \mid G = 1) = P(S = s \mid G = 2).$$

Our version of this constraint uses bins $B$ to empirically approximate the probability $P(S = s \mid G = g)$ and we also relax the equality to an $\epsilon$-approximate equality (for some pre-specified $\epsilon > 0$). Therefore, after transforming the scores using $\{x_{bb'}^{[g]}\}_{b,b'}$ the $\epsilon_{DP}$-approximate DP can be expressed via linear constraints[3]:

$$\left| \frac{1}{N^{[1]}} \sum_{b \in \mathcal{B}} x_{bb'}^{[1]} N_b^{[1]} - \frac{1}{N^{[2]}} \sum_{b \in \mathcal{B}} x_{bb'}^{[2]} N_b^{[2]} \right| \leq \epsilon_{DP} \qquad \forall \ b' \in \mathcal{B}, \tag{1}$$

where $N_b^{[g]}$ denote the number of observations from group $g$ in bin $b$ (before transformation), and $N^{[g]} = \sum_{b \in \mathcal{B}} N_b^{[g]}$. In practice, we choose $\epsilon_{DP}$ to be larger than the approximation error

---

[2]In applications, we can discretize the scores into $B$ bins with a quantile discretizer and consider how we can move them across bins. More bins allow for more granular interpretation of the transformed scores at the cost of us solving a harder problem and vice versa.

[3]For every $b'$, the constraint (1) can be written as a collection of linear constraints

$O(1/\sqrt{N^{[g]}})$ as we replace $P(S = s \mid G = g)$ with its empirical counterpart.

**Equalized Odds (EOdds):** The equalized odds condition for binary predictors, given in Hardt et al. 16, is a balance condition where the groups must have equal true positive and false positive rates. For continuous scores, it translates to having equal score distributions for each group conditional on their true labels [23]:

$$P(S = s \mid Y = y, G = 1) = P(S = s \mid Y = y, G = 2) \qquad \text{for } y \in \{0, 1\}.$$

Like demographic parity, our empirical score bin version requires that the distribution of positive or negative instances be $\epsilon_{EOdds}$-approximately equal between groups in the new bins $b'$. Both equal true positive rate and false positive rate can be expressed as linear constraints, respectively:

$$\left| \frac{1}{N_+^{[1]}} \sum_{b \in \mathcal{B}} x_{bb'}^{[1]} N_{b+}^{[1]} - \frac{1}{N_+^{[2]}} \sum_{b \in \mathcal{B}} x_{bb'}^{[2]} N_{b+}^{[2]} \right| \leq \epsilon_{EOdds} \ \forall \ b' \in \mathcal{B}$$

$$\left| \frac{1}{N_-^{[1]}} \sum_{b \in \mathcal{B}} x_{bb'}^{[1]} N_{b-}^{[1]} - \frac{1}{N_-^{[2]}} \sum_{b \in \mathcal{B}} x_{bb'}^{[2]} N_{b-}^{[2]} \right| \leq \epsilon_{EOdds} \ \forall \ b' \in \mathcal{B}. \tag{2}$$

**Predictive Rate Parity (PRP):** Lastly, we examine the predictive rate parity condition popularized in [10]. This condition states that the probability of being a positive instance is independent of group membership when we condition on the score. Formally:

$$P(Y = 1 \mid S = s, G = 1) = P(Y = 1 \mid S = s, G = 2).$$

Using the empirical score bin framework, an approximate version of the above implies that the proportion of positive instances in each bin must be $\epsilon_{PRP}$-approximately equal among groups:

$$\left| \frac{\sum_{b \in \mathcal{B}} x_{bb'}^{[1]} N_{b+}^{[1]}}{\sum_{b \in \mathcal{B}} x_{bb'}^{[1]} N_b^{[1]}} - \frac{\sum_{b \in \mathcal{B}} x_{bb'}^{[2]} N_{b+}^{[2]}}{\sum_{b \in \mathcal{B}} x_{bb'}^{[2]} N_b^{[2]}} \right| \leq \epsilon_{PRP} \ \forall \ b' \in \mathcal{B}. \tag{3}$$

Unlike constraints (1) and (2), which can be expressed as a linear function of the optimization variables $\{x_{bb'}^{[g]}\}$, condition (3) yields bilinear terms and is in general a non-convex constraint. A main technical difficulty of our framework arises from this non-convex fairness constraint—Section 3 presents an integer programming framework to handle this non-convexity, ensuring we can obtain a globally optimal solution to the resulting optimization problem.

*Remark.* Our definition of fairness as the worst-case violation across all bins aims to resemble approximations of the respective probabilistic definitions but we have not found identical definitions in other works. We comment on the differences and discuss why it does not contradict the traditional impossibility theorem of [18] in Appendix **??**.

## 2.2  MFOpt: Multiple Fairness Optimization Framework

We use constraints discussed in Section 2.1 to state the multiple fairness optimization (MFOpt) problem:

$$\underset{\{x_{bb'}^{[g]}\}_{b,b',g}}{\text{minimize}} \quad \sum_{g \in \mathcal{G}} \sum_{b \in \mathcal{B}} \sum_{b' \in \mathcal{B}} \left| \frac{N_b^{[g]}}{N} (\bar{s}_b - \bar{s}_{b'}) x_{bb'}^{[g]} \right| \tag{4a}$$

$$\text{s.t.} \quad \sum_{b \in \mathcal{B}} x_{bb'}^{[g]} = 1 \qquad \forall \ b' \in \mathcal{B}, \ g \in \mathcal{G} \tag{4b}$$

$$x_{bb}^{[g]} \geq 1 - \xi \qquad \forall \ b \in b' \in \mathcal{B}, \ g \in \mathcal{G} \tag{4c}$$

$$x_{bb'}^{[g]} = 0 \qquad \forall \ b' \ s.t. \ |b' - b| \geq w, \ \forall g \in \mathcal{G} \tag{4d}$$

$$\text{Fairness Constraints:} \quad (1), (2), (3) \tag{4e}$$

$$\frac{\sum_{b \in \mathcal{B}} x_{bb'}^{[g]} N_{b+}^{[g]}}{\sum_{b \in \mathcal{B}} x_{bb'}^{[g]} N_b^{[g]}} \leq \frac{\sum_{b \in \mathcal{B}} x_{b(b'+1)}^{[g]} N_{b+}^{[g]}}{\sum_{b \in \mathcal{B}} x_{b(b'+1)}^{[g]} N_b^{[g]}} \qquad \forall \ b' \in \{1, ..., B - 1\}, \ g \in \mathcal{G} \tag{4f}$$

$$0 \le x_{bb'}^{[g]} \le 1, \quad \forall \ b, b', g. \tag{4g}$$

Above, the optimization variables are the group-specific movement probability $x_{bb'}^{[g]}$ terms and all remaining terms are problem data and/or configurable hyperparameters. Starting with the objective (4a), we define $\bar{s}_b$ as the midpoint score in the bin and hence the objective is the product of the movement distance $\bar{s}_b - \bar{s}_{b'}$ weighed by the fraction of total samples moved $N_b^{[g]}/N$ and the amount of movement $x_{bb'}^{[g]}$. (4b) states that the total movement out of bin $b$, including the movement back to itself, must sum up to 1 and along with (4g) ensures that $x_{bb'}^{[g]}$ represent probabilities in a transition matrix. (4c) states that the total movement from bin $b$ back to itself must be lower bounded by hyperparameter $\xi$. This parameter controls how far we allow the new scores to stray from the original and is necessary to prevent zero denominators in 3 and 4f. Constraint (4d) represents window constraints to restrict extreme movements of scores beyond $w$ bins. Constraint (4e) are the fairness constraints in Section 2.1. (4f) ensures that we preserve the rank-ordering of the scores and expected values, which is desirable for comparing bins against each other. While constraints (4d), (4f) are not fairness-related, we add them to retain the utility of the solution. The predictive rate parity constraint (3) and (4f) both introduce non-convexities into Problem (4). These two constraints also require us to assume overlap, or that each bin contains at least one member from each group. Without overlap, predictive rate parity is undefined since it is not possible to compare expectations across groups for a given bin and demographic parity is likely violated as it means one group has zero probability of landing in a given bin.

Another option we considered for the objective following ([23]) is the Riemann approximation of AUC from the bins. ROC AUC be approximated by the FPR at bin $k$ and TPR of the cumulative bins $b \in \{k, \ldots, B\}$. Hence, it possible to optimize for maximizing AUC as a quadratic objective rather than minimizing score movement as a linear objective. However, in our empirical experience, maximizing AUC as the objective led to a harder numerical optimization problem (e.g, in terms of finding good feasible solutions) compared to minimizing score movement.

We close this section with two major benefits of this framework compared to inprocessing solutions such as adding fairness regularization ([32, 12, 27]) or using an entirely different fairness-based model ([8], [33]). First, the number of optimization variables scales in the order of $\mathcal{O}(|\mathcal{G}||\mathcal{B}|^2)$ while the number of constraints scale in the order of $\mathcal{O}(|\mathcal{B}||\mathcal{G}|^2)$. This is significant as it entails that our framework can be applied in arbitrarily large data settings as long as score-based binning is possible. Applying the transformation is equally scalable, as it only requires binning observations and making independent draws from a multinomial distribution with $B$ possible outcomes. Extending the framework to non-binary groups requires creating pairwise constraints for all groups in each bin to achieve the same fairness effect. This means that the number of constraints scale in the order of $\mathcal{O}(|\mathcal{B}||\mathcal{G}|^2)$ (in practice, we expect $|G|$ to be in the range of 3-4). This pairwise procedure does not affect the scaling of the number of optimization variables, which remains at $\mathcal{O}(|\mathcal{G}||\mathcal{B}|^2)$ [4].

The second benefit of our framework is that it returns a highly interpretable solution as it returns one optimized $B \times B$ transition matrix per group. Hence given a newly scored instance, several facts can be read from the corresponding row of the matrix such as the likelihood of moving to a specific bin $b$, moving into any higher or lower bin, etc. These probabilities can also be controlled via constraints as shown with the window constraints (4d) and max movement constraints (4c). This interpretability is an advantage over model regularization frameworks, where it is difficult to know how an individual's score might change when switching from the base model to a fair model or evaluating how much regularization is required to achieve a certain fairness effect.

## 3 Finding Optimal Solutions via Mixed Integer Programming (MIP)

The primary difficulty of the above optimization problem are the predictive rate parity constraint (3) and rank-order constraint (4f) which turn the problem non-convex. Non-convex constrained optimization is generally NP-hard and traditional methods that seek locally optimal solutions include gradient-based interior point optimization ([30]), sequential quadratic programming ([13]), or algo-

---

[4]Without the reformulation we present in 3, the pairwise constraints from pairwise groups would generate $\mathcal{O}(|\mathcal{B}|^3|\mathcal{G}|^2)$ bilinear terms from the cross-multiplication of the numerator/denominator sums and result in a much more computationally intensive problem.

rithms specific to quadratically-constrained-quadratic-programs (QCQPs) such as operator splitting methods, semidefinite relaxations, among others (see [24] for an overview).

Rather than pursuing a locally optimal solution, we study a reformulation ([7], [6]) of the problem as a mixed-integer-linear-program (MILP), which can theoretically be solved to global optimality in finite time via branch-and-bound methods [31]. Our reformulation grants two benefits over traditional locally optimal solvers. First, global strategies theoretically enable us to find the best possible solution. Second, as our problem primarily scales with the number of bins, we find that in practice, our problem is solvable to near-optimality when utilizing the power of modern MIP solvers.

## 3.1 Reformulations for computational efficiency

We first observe that a direct reformulation of the fractional terms into bilinear terms in constraints (3) and (4f) will lead to bilinear terms in the order of $\mathcal{O}(B^3)$. We show that we can reduce this to $\mathcal{O}(B)$ bilinear terms through a substitution that exploits the problem structure. Next, we take advantage of the vastly reduced number of bilinear terms to apply the normalized multiparametric disaggregation technique (NMDT, [1]) which we explain in Section 3.1.2. This allows us to approximate products of continuous variables as products of binary variables, which can be easily linearized and handled by MIP solvers. Importantly, this transformation of $xy$ (i.e, product) terms requires upper and lower bounds for $x$ and $y$ and we propose a method in Section 3.1.2 for generating and tightening these bounds by solving fractional linear program subproblems ([9]). Taken together, the reduction of bilinear terms combined with the bound-tightening procedure enable us to effectively apply the NMDT methodology and transform the problem from a non-convex QCQP to an MILP that can be solved to global optimality.

### 3.1.1 Step 1: Reducing the number of bilinear terms

We reduce the number of bilinear terms in our problem by making a substitution for the fraction term by introducing new optimization variables $v_b^{[g]} \geq 0$ (to represent a sum) and $t_b^{[g]} \geq 0$ (to represent the fractional quantity) as additional variables (see Appendix **??** for more details). We can then use them to write equivalent constraints with only $\mathcal{O}(B)$ bilinear terms.

Let,

$$v_{b'}^{[g]} = \sum_{b \in \mathcal{B}} x_{bb'}^{[g]} N_b^{[g]} \qquad \text{and} \qquad t_{b'}^{[g]} v_{b'}^{[g]} = \sum_{b \in \mathcal{B}} x_{bb'}^{[g]} N_{b+}^{[g]} \qquad \forall \ b' \in \mathcal{B}.$$

Then we have the following:

$$\text{Constraint (3)} \iff \left| t_{b'}^{[1]} - t_{b'}^{[2]} \right| \leq \epsilon_{PRP} \ \forall \ b' \in \mathcal{B},$$

$$\text{Constraint (4f)} \iff t_{b'}^{[g]} \leq t_{b'+1}^{[g]} \ \forall \ b' \in \{1, \dots, B-1\}$$

### 3.1.2 Step 2: NMDT and bound tightening through fractional LP subproblems

**Linearizing bilinear terms (NMDT):** We show how we can model each bilinear term $t_b^{[g]} v_b^{[g]}$ by using a binary expansion for the continuous variables $t_b^{[g]}, v_b^{[g]}$, and by observing that the product of binary variables can be modeled via integer programming (see [20, 29] for reference). To this end, we make use of the NMDT transformation [1]. We review this method below following [1][5] Given any bounded optimization variable $x \in [x_L, x_U]$, and precision factor $p$, a negative integer, we can represent this variable exactly as $x = (x_U - x_L)\lambda + x_L$ where

$$\lambda = \sum_{l \in \{-p, \dots, -1\}} 2^l z_l + \Delta\lambda$$

where $0 \leq \Delta\lambda \leq 2^p$ is a remainder term and $z_l \in \{0, 1\}$ are binary optimization variables. Dropping the remainder term $\Delta\lambda$ gives us the approximate form and product forms of $xy$ become dot products of several integer variables, which can be effectively handled via modern MIP solvers, such as [15]. Although we are solving an approximation (e.g. precision of $1e^{-4}$) this is not a practical problem

---

[5]The open-source implementation can be found in `https://github.com/joaquimg/` `QuadraticToBinary.jl` (MIT License) which we utilize.

since it is precise enough for reasonable choices of $\epsilon$ and we do not expect the constraints to hold exactly when we apply the post-processor on the testing data.

**Bounds on $v_b^{[g]}, t_b^{[g]}$ via Fractional LPs:** A key requirement to apply NMDT is that all optimization variables in the bilinear terms ($t_b^{[g]}$ and $v_b^{[g]}$ in our case) must be bounded and we need to be able to accurately estimate these bounds (i.e., tighter bounds leads to faster runtimes [1]). We propose obtaining these lower and upper by minimizing and maximizing respectively, the sub-problems while keeping all fairness constraints except the quadratic constraints. Firstly, note that bounds on $v_b^{[g]}$ can be solved as a simple LP (details in Appendix **??**). Meanwhile, obtaining a bound on $t_b^{[g]}$ requires solving a nonlinear problem due to fact that $t_b^{[g]}$ represents a fractional objective (ratio of two affine terms of optimization variables). However, we observe that we can apply the Charnes-Cooper transformation [9] to reformulate the nonlinear problem into a simple LP (details in Appendix **??**).

### 3.2 Choice of algorithm: QCQP (heuristic) vs MIP (optimal solution)

In this section we show the results and benefits of our reformulation from a non-convex QCQP to an MILP. In Table 1, we take each dataset, create a 60/40 train-test split, train a grid-searched random forest model, and score the training data. Next, we discretize the scores into bins, parameterize the problem (# bins, $\epsilon$, max movement, window size, solve time) on the scored training data, and compare our MILP solution solved by Gurobi ([15]) against the QCQP problem solved by IPOPT ([21]), which is a generic interior-point log-barrier penalty method for nonlinear constrained optimization. We discuss the datasets and problem parameters for all experiments in the Appendix **??**. For each metric such as $AUC$, we use $AUC_{INT}$ and $AUC_{IP}$ to denote the average result of applying the interior-point (INT, for short) or integer programming (IP) method, respectively. The metrics used are the objective value, optimality gap (%$\Delta$), [6] and $AUC$. We also report the statistical significance of the improvement based the $p$-value from the Wilcoxon signed-rank test to determine if %$\Delta_{IP} \leq$ %$\Delta_{INT}$ is a consistent result. Bold figures indicate statistical significance w.r.t. 1 standard deviation[7] [8].

Table 1: Interior Point Solution vs. MIP Solution

| Dataset | $Obj_{INT}$ | $Obj_{IP}$ | %$\Delta_{INT}$ | %$\Delta_{IP}$ | $p$-value | $AUC_{INT}$ | $AUC_{IP}$ |
|---|---|---|---|---|---|---|---|
| ACS Income | 2.0809 | 1.9682 | 15.076 ± 6.461 | 10.621 ± 3.402 | **0.0029** | 0.9041 | 0.9044 |
| ACS Insurance | 0.9769 | 0.9599 | 3.432 ± 0.225 | **1.715 ± 0.169** | **0.0010** | 0.7411 | 0.7413 |
| ACS Mobility | 2.4580 | 2.3781 | 5.37 ± 0.803 | **2.193 ± 0.138** | **0.0010** | 0.7971 | 0.7973 |
| ACS Poverty | 2.0693 | 2.0526 | 3.756 ± 0.435 | **2.972 ± 0.324** | **0.0010** | 0.8440 | 0.8440 |
| ACS Coverage | 8.9361 | 1.9665 | 79.711 ± 0.782 | **7.878 ± 2.207** | **0.0010** | 0.5420 | 0.8149 |
| ACS Travel | 2.3935 | 2.3859 | 2.554 ± 0.254 | 2.242 ± 0.28 | **0.0010** | 0.7725 | 0.7725 |
| Heart Disease | 1.8871 | 1.3035 | 26.385 ± 17.401 | **3.81 ± 0.864** | **0.0010** | 0.8302 | 0.8629 |
| COMPAS | 7.4551 | 3.1300 | 62.88 ± 13.407 | **17.055 ± 7.482** | **0.0010** | 0.5143 | 0.7378 |

As the results show, the MIP reformulation consistently beats the interior point solver applied on the raw optimization problem, even when solving for only 10 minutes. We also observe that in most cases, regardless of the method we choose, we can quickly find near optimal solutions that are high performing in the AUC sense. This is a significant result as it means that even a locally optimal solution to our optimization problem can yield a practically useful post-processing result. Additional experiments on these datasets showing the effectiveness of MFOpt with respect to performance and fairness in a training/testing data scenario can be found in Appendix **??**.

We conclude this section by reiterating two benefits of the reformulation. First, solving a MIP method yields lower bounds that can be used to prove optimality or otherwise gauge the quality of a feasible solution. Second, by framing the problem as a MIP, we can always theoretically continue improving

---

[6]The optimality gap is defined %$\Delta = \frac{UpperBound - LowerBound}{UpperBound}$ where the upper bound is the best feasible solution and the lower bound is produced by the branch-and-bound method. The INT method does not have the benefit of providing lower bounds, hence we use the bound produced by the IP method to compute this.

[7]We are limited in the number of trials we can run and actively chose to prioritize the variety of datasets we apply on rather than a large number of trials for a single dataset. As such, we expect relatively large standard errors but we reflect the consistency of our method through the p-value.

[8]See Appendix **??** for an explanation of the stark underperformance of IPOPT on the ACS Coverage and COMPAS datasets.

the solution to optimality based on the acceptable time limits. There are other applicable global optimization methods, such as spatial branch and bound. We considered these solutions but ultimately opted for a MIP approach due to the maturity and availability of solvers (see Appendix **??** for details).

## 4 Applications

In this section, we illustrate a few methods of applying our framework and how it can be used to help model developers select and understand models from a fairness perspective. To apply these procedures, we first require developing an efficient frontier of fairness solutions to understand which $\epsilon$ configurations are feasible for a given model type and dataset. To generate this frontier, we solve the problem over a grid of parameters $\epsilon_{DP}, \epsilon_{EOdds}, \epsilon_{PRP}$. Each feasible solution will yield a point $s \in \{(AUC, \epsilon_{DP}, \epsilon_{EOdds}, \epsilon_{PRP})\}$ and the collection of non-dominated points from the solution set yields a efficient frontier. We show the 2-d profile shots of our 4-d fairness surface in Figure 1 as an example, where the axes represent one of three fairness metrics and the point color gradient represents AUC. In theory, we could obtain a true Pareto-optimal frontier since we have devised a method of obtaining globally optimal solutions. However, we generate this frontier using IPOPT due to practical limitations as we are solving a $7 \times 7 \times 7$ grid of $\epsilon$ parameters.

Figure 1: Efficient frontier of solutions for ACS West Insurance data

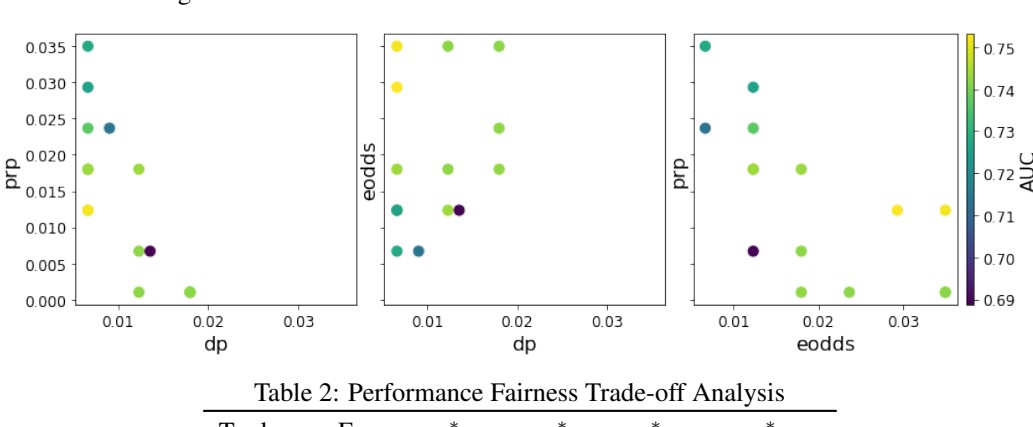

Table 2: Performance Fairness Trade-off Analysis

| Trade... | For... | $s^*_{AUC}$ | $s^*_{\epsilon_{DP}}$ | $s^*_{\epsilon_{EOdds}}$ | $s^*_{\epsilon_{PRP}}$ |
|---|---|---|---|---|---|
| Base | Base | 0.7434 | 0.0123 | 0.018 | 0.0123 |
| AUC | $\epsilon_{DP}$ | - | - | - | - |
| AUC | $\epsilon_{Eodds}$ | - | - | - | - |
| AUC | $\epsilon_{PRP}$ | 0.7422 | 0.0123 | 0.0180 | 0.0067 |
| $\epsilon_{DP}$ | $\epsilon_{PRP}$ | 0.7436 | 0.0152 | 0.0123 | 0.0067 |
| $\epsilon_{Eodds}$ | $\epsilon_{PRP}$ | - | - | - | - |
| $\epsilon_{Eodds}$ | $\epsilon_{DP}$ | 0.7436 | 0.0152 | 0.0123 | 0.006 |

### 4.1 Understanding fairness tradeoffs

After the Pareto frontier is generated, the modeler can pick an operating point $s$ based on the desired AUC and tolerable fairness violations $\epsilon$. However, when communicating fairness properties to stakeholders and accounting for potential changes in strategy, it can be helpful to additionally understand the cost of further increasing fairness in terms of $\epsilon_{DP}, \epsilon_{EOdds}, \epsilon_{PRP}$.

This tradeoff can be understood by looking at the characteristics of points on the frontier near the operating point. Suppose we are at an operating point $s$. If we want to trade $AUC$ for $\epsilon_{EOdds}$, then we would find a point $s'$ with at least as good $\epsilon_{DP}, \epsilon_{PRP}$ but worse $AUC$ and better $\epsilon_{EOdds}$. More generally, if we pick trade performance/fairness characteristic $c$ (cost) for characteristic $b$ (benefit), then we hold all other factors constant and find a point with better $b$ and worse $c$. We illustrate this in the Table 2. The first row shows a hypothetical operating point while the following rows show other points on the efficient frontier that we could move to when we make a certain trade. Blank rows indicate that no such point was found that permits the desired trade-off. This trade-off perspective can enable developers to better understand and communicate the costs to performance or other fairness metrics when trying to close the disparity in one fairness metric.

Table 3: Comparison with other fairness methods

| Method | Metric | Base | Testing Data Method | MF-Opt |
|---|---|---|---|---|
| Rezaei | $AUC$ | $0.7471 \pm 0.003$ | $0.6619 \pm 0.0022$ | $0.747 \pm 0.003$ |
| | $\epsilon_{DP}$ | $0.0117 \pm 0.0014$ | $0.0124 \pm 0.0013$ | $\mathbf{0.0088 \pm 0.001}$ |
| | $\epsilon_{EOdds}$ | $0.0266 \pm 0.007$ | $0.0291 \pm 0.0059$ | $\mathbf{0.0167 \pm 0.0029}$ |
| | $\epsilon_{PRP}$ | $0.109 \pm 0.0145$ | $0.1091 \pm 0.0143$ | $0.0986 \pm 0.0133$ |
| Pleiss | $AUC$ | $0.8319 \pm 0.0033$ | $0.8149 \pm 0.0087$ | $0.831 \pm 0.0032$ |
| | $\epsilon_{DP}$ | $0.0212 \pm 0.0016$ | $0.0137 \pm 0.0016$ | $\mathbf{0.0106 \pm 0.0011}$ |
| | $\epsilon_{EOdds}$ | $0.0329 \pm 0.0042$ | $0.023 \pm 0.0038$ | $\mathbf{0.0142 \pm 0.0028}$ |
| | $\epsilon_{PRP}$ | $0.1465 \pm 0.0178$ | $0.4147 \pm 0.1537$ | $0.1547 \pm 0.0293$ |

## 4.2 Performance Comparison

Lastly, we compare our framework against two methods and show that we can satisfy fairness constraint(s) just as well or better, while obtaining significantly stronger performance. Our comparison is done as follows (details in **??**), in each iteration we randomly split and pre-process the data, tune and fit a random forest model, and score the training and testing data to get the base scores $\hat{y}_0$. Next, we run the methodology that we are comparing against (i.e. build a model or apply the postprocessor) to get method scores $\hat{y}_m$. We then bin the outputs of the base model and compared method and compute the AUC along with the fairness metrics ($\epsilon_0, \epsilon_m$). Next, we solve our constrained optimization problem on the training data, where we set the parameters $\epsilon$ to $\frac{1}{2}min(\epsilon_0, \epsilon_m)$. After optimizing, we can apply the optimal solutions $x_{bb'}^{[g]}$ to assign new bins in the testing data based on the original score bins. One method of assigning a group $g$ instance with score $s \in b$ (denote as $s_b^{[g]}$) would be to randomly draw from a multinomial distribution parameterized by probabilities $(x_{b1}^{[g]}, x_{b2}^{[g]}, \ldots, x_{b|\mathcal{B}|}^{[g]})$. We propose alternative methods to this stochastic assignment method in the Appendix **??**. Lastly, we compute the resulting AUC and fairness metrics on remapped bins for the testing data (and do the same for $\hat{y}_0$ and $\hat{y}_m$ on the testing data). These figures are shown in Table 3. We only show the results on the test set due to space constraints and have placed the results for the training set in Appendix **??**. Bold figures indicate that a metric is statistically significant to 1 standard deviation.

First, we compare our framework against the in-processing framework in Rezaei et al. [27], which is a robust optimization-based logistic regression model for reducing equality of opportunity violation. We found that this method works well compared to standard logistic regression and managed to decrease fairness violations while maintaining the similar performance. It improves in $\epsilon_{Eodds}$ compared to a random forest as well. However, its weakness is that the underlying model is still logistic regression and therefore has limited expressiveness. In comparison, MFOpt can be applied on top of any model class and thus the performance advantages of more flexible models are better maintained.

Next, we compare our framework against the post-processing framework in Pleiss et al. [26]. In this method, the authors use randomization to maintain the model's calibration while simultaneously satisfying a relaxed equalized odds condition (whereby a linear combination of TPR and TNR are satisfied). Again, we see that this method maintains close performance as the base model, successfully shrinks equalized odds violations, and even decreases demographic parity violations too. However, it results in large violations of bin-wise predictive rate parity in contrast to our method.

## 4.3 Who's the fairest of them all?

In industry, machine learning model selection is guided by many factors including performance, speed, interpretability, among others. Yet, the fairness dimension is commonly overlooked unless the developers specifically induce it in their model. Even then, picking a specific fairness metric to optimize for can be a nebulous task. Rather than focus on a single metric, we describe a simple and intuitive method of gauging a model's efficiency in trading between different fairness definitions.

To do so, we first construct the frontier for the two models in question and then filter all points on the efficient frontier with tolerable performance $AUC \geq AUC_{min}$. Next, we find the point on the

respective frontiers with minimum Euclidean distance to the origin. The model with the shorter distance to their frontier can then be declared as the model that has better tradeoff properties. This procedure combined with the trade-off analysis can be useful when a developer is iterating between models that were not designed for fairness, but wants a model that can be flexibly made more fair through the MFOpt framework. As fairness requirements may change over time, the model that can yield the best tradeoffs between different definitions can offer the most overall utility.

# 5 Discussion

In our study, we have devised a flexible and interpretable post-processing method which we apply to push the limits of the impossibility theorem. We show that while theoretical limitations remain undisputed, there is a path forward to practically reconciling the conflicting fairness definitions. These results extend the findings of [28], which state that the trade-off of fairness and accuracy are negligible in practice. Our work reinforces this claim but also adds on that trade-offs between fairness definitions can be negligible as well. For further research, one area is to improve the consistency of the PRP violation reduction, as we observed the largest standard error in reducing this metric. This could be due to us using random forests for all experiments, which is known to be an uncalibrated model [4]. One method to address this is therefore first calibrating the model through other methodologies such as Platt scaling ([25]) or binning-based calibration ([19]) before applying MFOpt. We could also consider incorporating uncertainty in the training data through principled approaches such as stochastic or robust optimization.

# Acknowledgements

We would sincerely like to thank David Durfee and Osonde Osoba for their insightful feedback. We would also like to thank Ye Tu, Sakshi Jain, Shaunak Chatterjee, Ram Swaminathan, Romer Rosales, and Ya Xu for their support. Finally, we would also like to thank the anonymous reviewers for their helpful comments which improved the paper.

Rahul Mazumder contributed to this work while he was a consultant for LinkedIn (in compliance with MIT's outside professional activities policies). This work is not a part of his MIT research.

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
