# OpenReview forum: "Pushing the limits of fairness impossibility: Who's the fairest of them all?"
_NeurIPS.cc/2022/Conference — NeurIPS 2022 Accept_

### Official Review · Reviewer_mnAD · 2022-07-04

**Rating:** 7
**Confidence:** 3
**Soundness:** 3 good
**Presentation:** 3 good
**Contribution:** 3 good

**Summary:**

This paper studies how to deal with the trade-off among three common fairness notions: demographic parity, equalized odds, and predictive rate parity. The paper proposes a post-processing approach that aims to modify the decision made by any prediction model. The variable is the probability of flipping the decision from one to another for each group (in the case of continuous decision it is to move instances from one to another). Then, a combinatorial optimization problem is defined which contains the formulations of all fair constraints. The paper develops an efficient approach to solve the problem via mixed integer programming. Finally, experiments show that the proposed method can help the decision maker balance the trade-off between model utility and different fairness constraints.

**Questions:**

A critical question that may affect my final rating is that, in Eq. (1), when the number of instances in b’ after transformation is computed, why only considering the instances that are transformed into b’? What about the instances that are transformed out of b’?

Other questions:

Mixed integer programming in general is still NP-hard. Why does the paper claim that the proposed MIP reformulation is tractable?

The paper states that the challenge for solving the optimization problem comes from Constraint (3) and (4f) which introduce non-convexities. What if we remove Constraint (3) and (4f)? For Constraint (3), since the most commonly used fairness notions are demographic parity and equalized odds, in some cases it is fine not to consider the third one. For Constraint (4f), it seems to me that its purpose is to further preserve the utility of the solution in addition to the objective function. Thus, it is not a necessary condition in the optimization.


**Limitations:**

Yes.

**Strengths And Weaknesses:**

Strength

The proposed optimization framework is not limited to binary classification. The reformulation makes the optimization problem tractable. The optimization framework is also scalable since its complexity does not depend on the number of features and samples in the data. The experiment results are interesting which show that the proposed method is practically meaningful to practitioners and decision makers.

Weakness

Although the size the optimization problem only depends on the number of groups and the number of bins, in many cases the number of groups may be much larger than two. For example, when we consider intersectional fairness for multiple protected features, and when we consider conditional fairness notions (e.g., conditional demographic parity) which can produce a large number of subgroups by conditioning on certain features, the number of groups will increase significantly. The paper does not analyze the scalability of the proposed method under such situation.

---

> ### Author Response · Authors · 2022-08-01
> **Author responses to reviewer 3**
>
> Thank you for the detailed read of our paper and for your helpful comments in the review. We will first address your questions and then comment on the noted weaknesses and limitations.
>
> $\textbf{Questions:}$
>
> To first address the critical question regarding bin-wise constraints defined on $b’ \in \bf{B}$, we are indeed considering movement out of the respective $b’$. This is because $b’$ is included in the summation $\sum_{b \in B}x_{bb'}N_{bb'}$ and hence when we iterate over $x_{b’b’}$, which is the fraction of instances retained in the bin, having $x_{b’b’}$ strictly less than 1 it indicates that we only consider the fraction of instances that remain inside of the bin and discard the fraction that are moved out. To give an explicit example, suppose that we have 3 bins and are looking at the movement into bin 2, which is characterized by the vector $[0.1, 0.9, 0.2]$ (i.e., 10% of bin 1 moved into bin 2, 90% of bin 2 kept in bin 2, 20% of bin 3 moved to bin 2). This indicates that only 90% of bin 2 is retained and when we compute the term used in the constraint $x_{22}N_{2}$, this indeed accounts for the proportion that has been moved out. We hope this answers your primary concern.
>
> Regarding the tractability of MILP, you are certainly correct that it remains an NP-hard problem. We use “tractable” in the practical sense of being able to obtain a high-accuracy solution to the problem within a reasonable degree of optimality (in terms of optimality gap) within reasonable time for the specific application. As we show in our experiments in Section 3.2 (table 1), we could frequently solve down to <10% optimality gap in 10 minutes, which we believe is a practical enough solution.
>
> Regarding the challenges of constraint (3) and (4f), it is a keen observation that indeed, the non-convexities would disappear and we would be left with a linear program. However, the key motivation for our work is to embrace the challenge of satisfying the three conflicting fairness constraints that comprise the “impossibility theorem.” Predictive rate parity is an important pillar of the “impossibility theorem” as it, along with equal opportunity, were the two conflicting definitions noted in the seminal fairness paper by Chouldechova in 2016. Moreover, our review of recent literature (see below) indicates that predictive rate parity still remains a practically important criteria. Constraint (4f) is indeed a constraint that preserves the utility of the solution (higher scores imply higher expected outcome) and not strictly necessary. We have added a sentence clarifying this in Section 2.2 of our revised paper.
>
> $\textbf{Weaknesses:}$
>
> Lastly, regarding the scalability under intersections of groups, we first note that the framework is generalizable in the sense that we use pairwise constraints to define fairness among two groups. Hence if we wanted fairness for non-binary groups such that there is $\epsilon$-fairness between any two, we could create all pairwise constraints for each bin to achieve the same effect. Scalability wise, the number of optimization variables will scale in the order of $\mathcal{O}(|\mathcal{G}||\mathcal{B}|^2)$ and the constraints will scale in the order of $\mathcal{O}(|\mathcal{G}|^2|\mathcal{B}|)$. This is a good question and we added a section, Appendix A, to address this in our revised paper.
>
> Recent literature focusing on predictive rate parity in practice:
> 1. Can Active Learning Preemptively Mitigate Fairness Issues? [Branchaud-Charron et al. 2021]
> 2. Evaluation of Fairness Trade-offs in Predicting Student Success [Lee et al. 2020]
> 3. On the Applicability of Machine Learning Fairness Notions [Makhlouf et al. 2021]

---

> > ### Comment · Reviewer_mnAD · 2022-08-09
> > **Comment**
> >
> > I would like to thank the authors for their responses which address most of my concerns. However, I find the use of “tractable” misleading as it usually means “theoretically tractable”. I suggest replacing this term with the detailed explanation given in the authors’ responses.

---

> > > ### Author Response · Authors · 2022-08-09
> > > **Response to comment**
> > >
> > > Thank you for reading our response - we are glad that we could address your concerns. If we have the opportunity to further adjust the paper for the camera-ready version, we will incorporate your suggestion.

---

### Official Review · Reviewer_qwXE · 2022-07-07

**Rating:** 7
**Confidence:** 4
**Soundness:** 3 good
**Presentation:** 4 excellent
**Contribution:** 4 excellent

**Summary:**

This paper proposes an optimization-based method to simultaneously reduce the gap in demographic parity, equalized odds, and predictive rate parity. The fairness problem is first formulated as a non-convex optimization task and then solved using mixed integer programming. Experimental results that compares the proposed method against two existing baselines are provided.

**Questions:**


1. What change should be on the target and constrains if protected attributes are non-binary?

2. In table 1, INT significantly under-performs IP for ACS coverage and COMPAS. What's the reason behind this gap for these two datasets?

3. It would be great if the authors could provide additional ablation experiment on the size of bins and see how B impacts the performance.

**Limitations:**

Please see the weakness in the previous section for major concern. Minor concerns are:

1. Figure 1 needs to be improved: clearer labels for x,y axis and legend.

2. The experiment set up seems unclear. How was the training/validation/test set split? At which subset the X matrix is learned?



**Strengths And Weaknesses:**

Strengths:
1. This paper provides a novel solution to the well-known impossibility theorem in algorithmic fairness. The motif of the paper is novel and very relevant to the fairness community.
2. One major strength of this paper is that this methodology enables the user to select the model from the frontier that offers the best trade-off among all fairness metrics and performance. This feature is very appealing to models that are deployed under multiple applications.
3. This paper is well-organized. All concepts are clearly explained. The authors provide sufficient intuition along with rigorous definitions.
4. The formulation of the constrained optimization problem fits well with the fairness application. The constraints are well-explained and intuitively make perfect sense.
5. The proposed MILP solution is novel and suitable to this application. MILP is computationally tractable and globally optimal (although the problem formulation accepts tolerance).

Weakness:
The major concern for this approach is on its generalizability. Will the transformation matrix (X) learned on a tune/train set transfer (with great fairness & accuracy performance) to the test set? How is the binning method impact the generalizability? Technically, one can make B -> infinity and learn a X that achieves (almost) perfect performance on both fairness and accuracy perspective. But this performance will not generalize to the unseen test set.

---

> ### Author Response · Authors · 2022-08-01
> **Author response to reviewer 2**
>
> Thank you for the positive feedback on our work and for your detailed review - we appreciate the constructive feedback and questions. We will first address your questions and then comment on the weaknesses and limitations.
>
> $\textbf{Questions:}$
>
> Regarding non-binary protected attributes, our methodology can indeed be generalized. Since the fairness constraints are expressed on a pairwise basis, we can achieve the same effect for $G$ groups by defining the pairwise fairness constraints across all (G choose 2)=G(G-1)/2 possible pairs. This is a good question and we have added additional commentary in Appendix A.
>
> Regarding the underperformance of the interior-point method in those two instances mentioned in Table 1 (COMPAS and ACS coverage): across the 10 trials we ran, the IPOPT algorithm had frequently failed to converge within the 10-minute time limit for these two datasets.  From the outputs, we saw that convergence failure is accompanied by heavy violation of the predictive rate parity constraint (demographic parity and equalized odds are still satisfied) and a high objective value. The exact reason for frequent failure in these two datasets is unclear, however, we hypothesize that it is due to a relatively high predictive rate parity gap in the data which led to numeric issues. Such failures were not observed in the IP formulation while both methods were provided the exact same data parameters.
>
> We have added Appendix I to document an ablation study regarding parameter B where we discuss that the specific choice of B is an important but challenging question and show some empirical results. The ablation study expectedly shows that as B increases, we tend to have higher resolution in the transformed scores, leading to stronger out-of-sample performance in terms of ROC and PR AUC. The performance also plateaus beyond a certain B, indicating that selecting a large B is not necessary to achieve good out-of-sample performance. The results are more ambiguous for out-of-sample fairness, where increasing B does not strictly lead to better generalization but also does not cause the metrics to fluctuate wildly. However, we caveat that comparing the fairness of solutions using different B is somewhat ill-defined and describe this in the ablation study. Another generalizability study can be found in Appendix H, where we show that using B=50 indeed generalizes well in a traditional training/testing setting, giving us very comparable performance and significantly better fairness on out-of-sample data.
>
> $\textbf{Weaknesses:}$
>
> Regarding the choice of B on a more theoretical basis, a rigorous analysis of the optimal number of bins to use presents challenges which we have deemed out of scope for our work. Like you mentioned, having more bins will lower the bias in estimating the score-transformation function at the cost of having high variance and low generalizability for both performance and fairness metrics.
>
> While it is clear that B should tend to infinity as the sample size tends to infinity for achieving asymptotic unbiasedness, we should additionally have $N/B \longrightarrow \infty$ for the generalizability. For example, if $B = N^{1/3}$ (where $N$ denotes the training sample size) and we do quantile-based binning, we would have $N^{2/3}$ samples in each bin (which would allow us to control the variance of the estimates in each bin). The derivation of optimal scaling of $B$ is an extremely challenging problem and is out of scope of our paper. However, we comment on the role of $B$ in the bias-variance trade-off and point out the limitation of this work in Appendix I along with the ablation study.
>
> Lastly, thank you for the note on the experiment setup and Figure 1. We appreciate your close attention to these details and the constructive feedback. We have added additional details regarding the experiments in the body and Appendix E, stating that we use a 60/40 train/test split via random sampling. We have also made Figure 1 more legible.

---

### Official Review · Reviewer_nPyX · 2022-07-10

**Rating:** 5
**Confidence:** 3
**Soundness:** 2 fair
**Presentation:** 2 fair
**Contribution:** 2 fair

**Summary:**

This paper pushes the limit of fairness impossibility theorem where all three group fairness definitions cannot be satisfied simultaneously. Based on an integer-programming approach, the authors propose a framework to yield a certifiably optimal post-processing method for simultaneously satisfying multiple fairness criteria under small violations. In experiments, the framework with minimal model performance reduction achieves fairness improvement across different definitions simultaneously. As an application, the framework advises on model selection and fairness explainability.

**Questions:**

- In Figure 1, what does different colours mean?
- Why do authors compare with an in-processing algorithm while the proposed work is a post-processing algorithm?

**Limitations:**

There is no negative societal impact of this work.

**Strengths And Weaknesses:**

Strength:
- The motivation of the paper is very interesting.

Weakness:
- There is no experimental comparison with existing fairness works.
 - While the proposed method belongs to post-processing fairness algorithms, experiments do not include any comparison with post-processing algorithms.
- Tables and figures are uninformatively presented.
- In Table 3, few cells are in bold fonts. No interpretation is given.
- A motivating example is missing, where authors could compare theoretical fairness impossibility result vs. their proposed solution.

---

> ### Author Response · Authors · 2022-08-01
> **Author response to reviewer 1**
>
> Thank you for your review and comments regarding the methodology comparisons and clarity of presentation. We hope our response clarifies your questions and concerns:
>
> $\textbf{Questions:}$
>
> We wish to clarify that for experimental comparisons with existing fairness works, we have indeed chosen two methodologies to compare against in Section 2 (pg. 9). Specifically:
> Rezaei et al.’s methodology (an in-processing solution based on regularized logistic regression that improves equalized odds)
> Pleiss et al.’s methodology (a post-processing solution that utilizes a randomized reranker that improves also improves equalized odds)
> We chose these methodologies after parsing through recent literature and checking the applicability of the algorithm and availability of implementation. For instance, we examined various implementations in the open-source package AIF360 (https://aif360.mybluemix.net/), but we did not select them for comparison as they were either inapplicable (made for binary {0,1} classifiers) or did not work well out-of-the-box, even after a substantial effort on our end. On top of comparing against a post-processing solution, we also compared our method against Razaei et al.’s in-processing method because post-processing methods are sometimes viewed as weaker than in-processing since it is not customized to the base modeling algorithm. However, we do not observe this weakness in our methods in the experiments in Table 3. Lastly, we note that on top of these comparisons, we have additional experiments in a classical train/test split setting in Appendix H. We hope this clarifies your main concern and we would really appreciate it if you revise your score accordingly.
>
> $\textbf{Weaknesses:}$
>
> Regarding the clarity of the presentation, we thank you for the feedback and have added details to clarify in the revised PDF. The bolded figures in Table 3 refer to 1-standard deviation statistical significance, which we have added to the description. The colors in figure 1 correspond to AUC and we have revised this figure. Thank you for the suggested improvements.

---

### Meta-Review · Area_Chair_37Nh · 2022-08-30

**Recommendation:** Accept
**Confidence:** Less certain

**Metareview:**

This paper provides an interesting framework that investigates trade-offs of multiple fairness criteria. The reviewers agree that the proposed techniques are interesting and make meaningful contributions to algorithmic fairness. Please incorporate the reviewers' suggestions in your next revision (e.g., not using "tractable" to describe MIP).


**Award:**

No

---

### Decision · Program_Chairs · 2022-09-14

Accept